# Density estimation from unweighted k-nearest neighbor graphs: a roadmap

**Ulrike von Luxburg**     and     **Morteza Alamgir**
Department of Computer Science
University of Hamburg, Germany
{luxburg,alamgir}@informatik.uni-hamburg.de

## Abstract

Consider an unweighted $k$-nearest neighbor graph on $n$ points that have been sampled i.i.d. from some unknown density $p$ on $\mathbb{R}^d$. We prove how one can estimate the density $p$ just from the unweighted adjacency matrix of the graph, without knowing the points themselves or any distance or similarity scores. The key insights are that local differences in link numbers can be used to estimate a local function of the gradient of $p$, and that integrating this function along shortest paths leads to an estimate of the underlying density.

## 1  Introduction

**The problem.** Consider an unweighted $k$-nearest neighbor graph that has been built on a random sample $X_1, ..., X_n$ from some unknown density $p$ on $\mathbb{R}^d$. Assume we are given the adjacency matrix of the graph, but we do *not* know the point locations $X_1, ...., X_n$ or any distance or similarity scores between the points. Is it then possible to estimate the underlying density $p$, just from the adjacency matrix of the unweighted graph?

**Why is this problem interesting for machine learning?** Machine learning algorithms on graphs are abundant, ranging from graph clustering methods such as spectral clustering over label propagation methods for semi-supervised learning to dimensionality reduction methods and manifold algorithms. In the majority of applications, the graphs that are used as input are similarity graphs: Given a set of abstract "objects" $X_1, ..., X_n$ we first compute pairwise similarities $s(X_i, X_j)$ according to some suitable similarity function and then build a $k$-nearest neighbor graph (kNN graph for short) based on this similarity function. The intuition is that the edges in the graph encode the local information given by the similarity function, whereas the graph as a whole reveals global properties of the data distribution such as cluster properties, high- and low-density regions, or manifold structure. From a computational point of view, kNN graphs are convenient because they lead to a sparse representation of the data — even more so when the graph is unweighted. From a statistical point of view the key question is whether this sparse representation still contains all the relevant information about the original data, in particular the information about the underlying data distribution. It is easy to see that for suitably *weighted* kNN graphs this is the case: the original density can be estimated from the degrees in the graph. However, it is completely unclear whether the same holds true for *unweighted* kNN graphs.

**Why is the problem difficult?**   The naive attempt to estimate the density from vertex degrees obviously has to fail in unweighted kNN graphs because all vertex degrees are (about) $k$. Moreover, unweighted kNN graphs are invariant with respect to rescaling of the underlying distribution by a constant factor (e.g., the unweighted kNN graph on a sample from the uniform distribution on $[0, 1]^2$ is indistinguishable from a kNN graph on a sample from the uniform distribution on $[0, 2]^2$). So all we can hope for is an estimate of the density up to some multiplicative constant that cannot be determined from the kNN graph alone. The main difficulty, however, is that a kNN graph "looks

the same" in every small neighborhood. To see this, consider the case where the underlying density is continuous, hence approximately constant in small neighborhoods. Then, if $n$ is large and $k/n$ is small, local neighborhoods in the kNN graph are all going to look like kNN graphs from a uniform distribution. This intuition raises an important issue. *It is impossible to estimate the density in an unweighted kNN graph by local quantities alone.* We somehow have to make use of global properties if we want to be successful. This makes the problem very different and much harder than more standard density estimation problems.

**Our solution.** We show that it is indeed possible to estimate the underlying density from an unweighted kNN graph. The construction is fairly involved. In a first step we estimate a pointwise function of the gradient of the density, and in a second step we integrate these estimates along shortest paths in the graph to end up with an approximation of the log-density. Our estimate works as long as the kNN graph is reasonably dense ($k^{d+2}/(n^2 \log^d n) \to \infty$). However, it fails in the more important sparser regime (e.g., $k \approx \log n$). Currently we do not know whether this is due to a suboptimal proof or whether density estimation is generally impossible in the sparse regime.

## 2 Notation and assumptions

**Underlying space.** Let $\mathcal{X} \subset \mathbb{R}^d$ be a compact subset of $\mathbb{R}^d$. Denote by $\partial \mathcal{X}$ the topological boundary of $\mathcal{X}$. For $\varepsilon > 0$ define the $\varepsilon$-interior $\mathcal{X}_\varepsilon := \{x \in \mathcal{X} \mid d(x, \partial \mathcal{X}) \geq \varepsilon\}$. We assume that $\mathcal{X}$ is "full dimensional" in the sense that there exists some $\varepsilon_0 > 0$ such that $\mathcal{X}_{\varepsilon_0}$ is non-empty and connected. By $\eta_d$ we denote the volume of a $d$-dimensional unit ball, and by $v_d$ the volume of the intersection of two $d$-dimensional unit balls whose centers have distance 1.

**Density.** Let $p$ be a continuously differentiable density on $\mathcal{X}$. We assume that there exist constants $p_{\min}$ and $p_{\max}$ such that $0 < p_{\min} \leq p(x) \leq p_{\max} < \infty$ for all $x \in \mathcal{X}$.

**Graph.** Given an i.i.d. sample $\mathcal{X}_n := \{X_1, ..., X_n\}$ from $p$, we build a graph $G_n = (V_n, E_n)$ with $V_n = \mathcal{X}_n$. We connect $X_i$ by a directed edge to $X_j$ if $X_j$ is among the $k$-nearest neighbors of $X_i$. The resulting graph is called the directed, unweighted kNN graph (in the following, we will often drop the words "directed" and "unweighted"). By $r(x) := r_{n,k}(x)$ we denote the Euclidean distance of a point $x$ to its $k$th nearest neighbor. For any vertex $x \in V$ we define the sets

$$\text{In}(x) := \text{In}_{n,k}(x) := \{y \in \mathcal{X}_n \mid (y, x) \in E_n\} \qquad \text{(source points of in-links to } x\text{)}$$

$$\text{Out}(x) := \text{Out}_{n,k}(x) := \{y \in \mathcal{X}_n \mid (x, y) \in E_n\} \qquad \text{(target points of out-links from } x\text{)}.$$

To increase readability we often omit the indices $n$ and $k$. For a finite set $S$ we denote by $|S|$ its number of elements.

**Paths.** For a rectifiable path $\gamma : [0, 1] \to \mathcal{X}$ we define its $p$-weighted length as

$$\ell_p(\gamma) := \int_\gamma p^{1/d}(x)\, ds \ := \int_0^1 p^{1/d}(\gamma(t))|\gamma'(t)|\, dt$$

(recall the notational convention of writing "$ds$" in a line integral). For two points $x, y \in \mathcal{X}$ we define their $p$-weighted distance as $D_p(x, y) = \inf_\gamma \ell_p(\gamma)$ where the infimum is taken over all rectifiable paths $\gamma$ that connect $x$ to $y$. As a consequence of the compactness of $\mathcal{X}$, a minimizing path that realizes $D_p$ always exists (cf. Burago et al., 2001, Section 2.5.2). We call such a path a $D_p$-shortest path. Under the given assumptions on $p$, the $D_p$-shortest path between any two points $x, y \in \mathcal{X}_{\varepsilon_0}$ is smooth.

In an unweighted graph, define the length of a path as its number of edges. For two vertices $x, y$ denote by $D_{sp}(x, y)$ their shortest path distance in the graph. It has been proved in Alamgir and von Luxburg (2012) that for unweighted, undirected kNN graphs, $(k/(n\eta_d))^{1/d} D_{sp}(x, y) \to D_p(x, y)$ almost surely as $n \to \infty$ and $k \to \infty$ appropriately slowly. The proofs extend directly to the case of directed kNN graphs.

## 3 Warmup: the 1-dimensional case

To gain some intuition about the problem and its solution, let us consider the 1-dimensional case $\mathcal{X} \subset \mathbb{R}$. For any given point $x \in \mathcal{X}_n$ we define the following sets:

$$\text{Left}_1(x) := |\{y \in \text{Out}(x) \mid y < x\}| \qquad \text{and} \qquad \text{Right}_1(x) := |\{y \in \text{Out}(x) \mid y > x\}|.$$

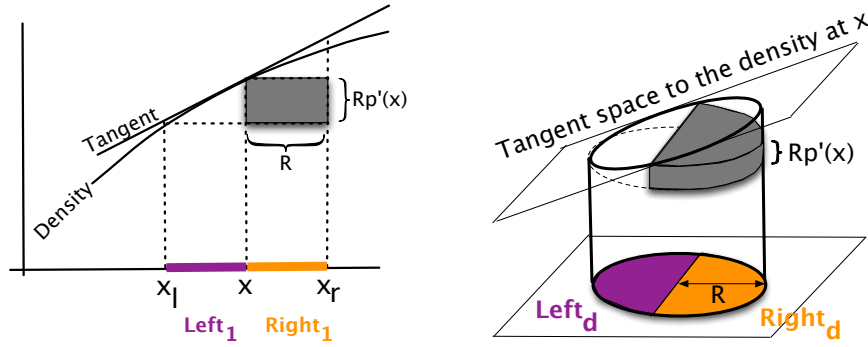

Figure 1: Geometric argument (left: 1-dimensional case, right: 2-dimensional case). The difference $\text{Right} - \text{Left}$ is approximately proportional to the volume of the grey-shaded area.

The intuition to estimate the density from the directed kNN graph is the following. Consider a point $x$ in a region where the density has positive slope. The set $\text{Out}(x)$ is approximately symmetric around $x$, that is it has the form $\text{Out}(x) = \mathcal{X}_n \cap [x - R, x + R]$ for some $R > 0$. When the density has an increasing slope at $x$, there tend to be less sample points in $[x - R, x]$ than in $[x, x + R]$, so the set $\text{Right}_1(x)$ tends to contain more sample points than the set $\text{Left}_1(x)$. This is the effect we want to exploit. The difference $\text{Right}_1(x) - \text{Left}_1(x)$ can be approximated by $n \cdot (P([x, x+R]) - P([x - R, x]))$, and by a simple geometric argument one can see that the latter probability is approximately $R^2 p'(x)$. See Figure 1 (left side) for an illustration. By standard concentration arguments one can see that if $n$ is large enough and $k$ chosen appropriately, then $R \approx k/(2np(x))$. Plugging these two things together shows that $\text{Right}_1(x) - \text{Left}_1(x) \approx (k^2/(4n^2)) \cdot p'(x)/p^2$, hence gives an estimate of $p'(x)/p^2(x)$. But we are not there yet: it is impossible to directly turn an estimate of $p'(x)/p^2(x)$ into an estimate of $p(x)$. This is in accordance with the intuition we mentioned above: one cannot estimate the density by just looking at a local neighborhood of $x$ in the kNN graph. Here is now the key trick to introduce a global component to the estimate. We fix one data point $X_0$ that is going to play the role of an anchor point. To estimate the density at a particular data point $X_s$, we now sum the estimates $p'(x)/p^2(x)$ over all data points $x$ that sit between $X_0$ and $X_s$. This corresponds to integrating the function $p'(x)/p^2(x)$ over the interval $[X_0, X_s]$ with respect to the underlying density $p$, which in turn corresponds to integrating the function $p'(x)/p(x)$ over the interval $[X_0, X_s]$ with respect to the standard Lebesgue measure. This latter integral is well known, its primitive is $\log p(x)$. Hence, for each data point $X_s$ we get an estimate of $\log p(X_s) - \log p(X_0)$. Then we exponentiate and arrive at an estimate of $c \cdot p(x)$, where $c = 1/p(X_0)$ plays the role of an unknown constant.

## 4    A hypothetical estimate in the $d$-dimensional case

We now generalize our approach to the $d$-dimensional setting. There are two main challenges: First, we need to replace the integral over all sample points between $X_0$ and $X_s$ by something more general in $\mathbb{R}^d$. Our idea is to consider an integral along a path between $X_0$ and $X_s$, specifically along a path that corresponds to a shortest path in the graph $G_n$. Second, we need a generalization of the concept of what are "left" and "right" out-links. Our idea is to use the shortest path as reference. For a point $x$ on the shortest path between $X_0$ and $X_s$, the "left points" of $x$ should be the ones that are on or close to the subpath from $X_0$ to $x$ and "right points" the ones on or close to the path from $x$ to $X_s$.

### 4.1    Gradient estimates based on link differences

Fix a point $x$ on a simple, continuously differentiable path $\gamma$ and let $T(x)$ be its tangent vector. Consider $h(y) = \langle w, y \rangle + b$ with normal vector $w := T(x)$, where the offset $b$ has been chosen such that the hyperplane $H := \{y \in \mathbb{R}^d \mid h(y) = 0\}$ goes through $x$. Define

$$\text{Left}_d(x) := \text{Left}_{d,n,k}(x) := |\{x \in \text{Out}(x) \mid h(x) \leq 0\}|$$

$$\text{Right}_d(x) := \text{Right}_{d,n,k}(x) := |\{x \in \text{Out}(x) \mid h(x) > 0\}|.$$

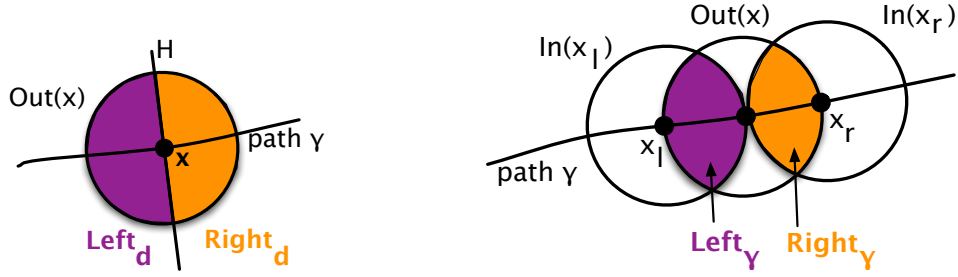

Figure 2: Definitions of "left" and "right"in the $d$-dimensional case.

See Figure 2 (left side) for an illustration. This definition is a direct generalization of the definition of $\text{Left}_1$ und $\text{Right}_1$ in the 1-dimensional case. It is not yet the end of the story, as the quantities $\text{Left}_d$ and $\text{Right}_d$ cannot be evaluated based on the kNN graph alone, but it is a good starting point to develop the necessary proof concepts. In this section we prove the consistency of a density estimate based on $\text{Left}_d$ and $\text{Right}_d$. In Section 5 we will further generalize the definition to our final estimate.

**Theorem 1 (Estimate related to the gradient)** *Let $\mathcal{X}$ and $p$ satisfy the assumptions in Section 2. Let $\gamma$ be a differentiable, regular, simple path in $\mathcal{X}_{\varepsilon_0}$ and $x$ a sample point on this path. Let $T$ be the tangent direction of $\gamma$ at $x$ and $p'_T(x)$ the directional derivative of the density $p$ in direction $T$ at point $x$. Then, if $n \to \infty$, $k \to \infty$, $k/n \to 0$, $k^{d+2}/n^2 \to \infty$,*

$$\frac{2n^{1/d}\eta_d^{1/d}}{k^{(d+1)/d}}\Big(Right_{d,n,k}(x) - Left_{d,n,k}(x)\Big) \longrightarrow \frac{p'_T(x)}{p(x)^{(d+1)/d}} \quad a.s.$$

*If $k^{d+2}/(n^2 \log^d n) \to \infty$ the convergence even holds uniformly over all sample points $x \in \mathcal{X}_n$.*

**Proof sketch.** The key problem in the proof is that the difference $\text{Right}_d - \text{Left}_d$ is of a much smaller order of magnitude than $\text{Right}_d$ and $\text{Left}_d$ themselves, so controlling the deviations of $\text{Right}_d - \text{Left}_d$ is somewhat tricky. Conditioned on $r_{out}(x) =: r$, $\text{Right}_d \sim Bin(k, \pi_r)$ and $\text{Left}_d \sim Bin(k, \pi_l)$, where $\pi_r = P(\text{right half ball})/P(\text{ball})$ and $\pi_l$ analogously (cf. Figure 2). By Hoeffding's inequality, $\text{Right}_d - \text{Left}_d \approx E(\text{Right}_d - \text{Left}_d) \pm \Theta(\sqrt{k})$ with high probability. Note that $\pi_l$ and $\pi_r$ tend to be close to $1/2$, thus Hoeffding's inequality is reasonably tight. A simple geometric argument shows that if the density in a neighborhood of $x$ is linear, then $E(\text{Right}_d - \text{Left}_d) = n \cdot r^d \eta_d/2 \cdot r p'_T(x)$ ($n$ times the probability mass of the grey area in Figure 1). A similar argument holds approximately if the density is just differentiable at $x$. A standard concentration argument for the out-radius shows that with high probability, $r_{out}(x)$ can be approximated by $(k/(n\eta_d p(x)))^{1/d}$. Combining all results we obtain that with high probability,

$$\frac{2n^{1/d}\eta_d^{1/d}}{k^{(d+1)/d}}(\text{Right}_d - \text{Left}_d) = \frac{p'_T(x)}{p(x)^{(d+1)/d}} \pm \Theta\Big(\frac{n^{1/d}}{k^{1/2+1/d}}\Big).$$

Convergence takes place if the noise term on the right hand side goes to 0 and the "high probability" converges to 1, which happens under the conditions on $n$ and $k$ stated in the theorem. ☺

## 4.2 Integrating the gradient estimates along the shortest path

To deal with the integration part, let us recap some standard results about line integrals.

**Proposition 2 (Line integral)** *Let $\gamma : [0,1] \to \mathbb{R}^d$ be a simple, continuously differentiable path from $x_0 = \gamma(0)$ to $x_1 = \gamma(1)$ parameterized by arc length. For a point $x = \gamma(t)$ on the path, denote by $T(x)$ the tangent vector to $\gamma$ at $x$, and by $p'_T(x)$ the directional derivative of $p$ in the tangent direction $T$. Then*

$$\int_\gamma \frac{p'_T(x)}{p(x)} ds = \log(p(x_1)) - \log(p(x_0)).$$

**Proof.** We define the vector field

$$F : \mathbb{R}^d \to \mathbb{R}^d, \ x \mapsto \frac{p'(x)}{p(x)} = \frac{1}{p(x)} \begin{pmatrix} \partial p / \partial x_1 \\ \ldots \\ \partial p / \partial x_d \end{pmatrix}.$$

Observe that $F$ is a continuous gradient field with primitive $V : \mathbb{R}^d \to \mathbb{R}, \ x \mapsto \log(p(x))$. Now consider the line integral of $F$ along $\gamma$:

$$\int_\gamma F(x)\, dx \overset{\text{def}}{=} \int_0^1 \left\langle F(\gamma(t)), \gamma'(t) \right\rangle dt = \int_0^1 \frac{1}{p(\gamma(t))} \left\langle p'(\gamma(t)), \gamma'(t) \right\rangle dt. \tag{1}$$

Note that $\gamma'(t)$ is the tangent vector $T(x)$ of the path $\gamma$ at point $x = \gamma(t)$. Hence, the scalar product $\langle p'(\gamma(t)), \gamma'(t) \rangle$ coincides with the directional derivative of $p$ in direction $T$, so the right hand side of Equation (1) coincides with the left hand side of the equation in the proposition. On the other hand, it is well known that the line integral over a gradient field only depends on the starting and end point of $\gamma$ and is given by

$$\int_\gamma F(x)\, dx = V(x_1) - V(x_0).$$

This coincides with the right hand side of the equation in the proposition. ☺

Now we consider the finite sample case. The goal is to approximate the integral along the continuous path $\gamma$ by a sum along a path $\gamma_n$ in the kNN graph $G_n$. To achieve this, we need to construct a sequence of paths $\gamma_n$ in $G_n$ such that $\gamma_n$ converges to some well-defined path $\gamma$ in the underlying space and the lengths of $\gamma_n$ in $G_n$ converge to $\ell_p(\gamma)$. To this end, we are going to consider paths $\gamma_n$ which are shortest paths in the graph.

Adapting the proof of the convergence of shortest paths in unweighted kNN graphs (Alamgir and von Luxburg, 2012) we can derive the following statement for integrals along shortest paths.

**Proposition 3 (Integrating a function along a shortest path)** *Let $\mathcal{X}$ and $p$ satisfy the assumptions in Section 2. Fix two sample points in $\mathcal{X}_{\varepsilon_0}$, say $X_0$ and $X_s$, and let $\gamma_n$ be a shortest path between $X_0$ and $X_s$ in the kNN graph $G_n$. Let $\gamma \subset \mathcal{X}$ be a path that realizes $D_p(X_0, X_s)$. Assume that it is unique and is completely contained in $\mathcal{X}_{\varepsilon_0}$. Let $g : \mathcal{X} \to \mathbb{R}$ be a continuous function. Then, as $n \to \infty$, $k^{1+\alpha}/n \to 0$ (for some small $\alpha > 0$), $k/\log n \to \infty$,*

$$\left( \frac{k}{n \eta_d} \right)^{1/d} \cdot \sum_{x \in \gamma_n} g(x) \longrightarrow \int_\gamma g(x) p(x)^{1/d}\, ds \quad a.s.$$

Note that if $g(x) p^{1/d}(x)$ can be written in the form $\langle F(\gamma(t)), \gamma'(t) \rangle$, then the same statement even holds if the shortest $D_p$-path is not unique, because the path integral then only depends on start and end point. This is the case for our particular function of interest, $g(x) = p'_T(x)/p^{1+1/d}(x)$.

### 4.3  Combining everything to obtain a density estimate

**Theorem 4 (Density estimate)** *Let $\mathcal{X}$ and $p$ satisfy the assumptions in Section 2, let $X_0 \in \mathcal{X}_{\varepsilon_0}$ be any fixed sample point. For another sample point $X_s$, let $\gamma_n$ be a shortest path between $X_0$ and $X_s$ in the kNN graph $G_n$. Assume that there exists a path $\gamma$ that realizes $D_p(x, y)$ and that is completely contained in $\mathcal{X}_{\varepsilon_0}$. Then, as $n \to \infty$, $k \to \infty$, $k/n \to 0$, $k^{d+2}/(n^2 \log^d n) \to \infty$,*

$$\frac{2}{k} \sum_{x \in \gamma_n} \left( \text{Right}_{d,n,k}(x) - \text{Left}_{d,n,k}(x) \right) \longrightarrow \log p(X_s) - \log p(X_0) \quad a.s.$$

**Proof sketch.** By Proposition 2,

$$\log(p(X_s)) - \log(p(X_0)) = \int_\gamma \frac{p'_T(x)}{p(x)}\, ds = \int_\gamma \frac{p'_T(x)}{p(x)^{(d+1)/d}} p(x)^{1/d}\, ds.$$

According to Proposition 3, the latter can be approximated by

$$\left(\frac{k}{n\eta_d}\right)^{1/d} \sum_{x \in \gamma_n} \frac{p_T'(x)}{p(x)^{(d+1)/d}}$$

where $\gamma_n$ is a shortest path between $X_0$ and $X_s$ in the kNN graph. Proposition 1 shows that this quantity gets estimated by

$$\left(\frac{k}{n\eta_d}\right)^{1/d} \frac{n^{1/d}}{k^{(d+1)/d}} \cdot 2\eta_d^{1/d} \sum_{x \in \gamma_n} \left(\text{Right}_d(x) - \text{Left}_d(x)\right) = \frac{2}{k} \sum_{x \in \gamma_n} \left(\text{Right}_d(x) - \text{Left}_d(x)\right).$$

☺

## 5    The final $d$-dimensional density estimate

In this section, we finally introduce an estimate that solely uses quantities available from the kNN graph. Let $x$ be a vertex on a shortest path $\gamma_{n,k}$ in the kNN graph $G_n$. Let $x_l$ and $x_r$ be the predecessor and successor vertices of $x$ on this path (in particular, $x_l$ and $x_r$ are sample points as well). Define

$$\text{Left}_{\gamma_{n,k}}(x) := |\operatorname{Out}(x) \cap \operatorname{In}(x_l)| \qquad \text{and} \qquad \text{Right}_{\gamma_{n,k}}(x) := |\operatorname{Out}(x) \cap \operatorname{In}(x_r)|.$$

See Figure 2 (right side) for an illustration. On first glance, these sets look quite different from $\text{Left}_d$ and $\text{Right}_d$. But the intuition is that whenever we find two sets on the left and right side of $x$ that have approximately the same volume, then the difference $\text{Left}_{\gamma_{n,k}} - \text{Right}_{\gamma_{n,k}}$ should be a function of $p_T'(x)$. For a second intuition consider the special case $d = 1$ and recall the definition of $R$ of Section 3. One can show that in expectation, $[x - R, x]$ coincides with $\operatorname{Out}(x) \cap \operatorname{In}(x_l)$ and $[x, x + R]$ with $\operatorname{Out}(x) \cap \operatorname{In}(x_r)$, so in case $d = 1$ the definitions coincide in expectation with the ones in Section 3. Another insight is that the set $\text{Left}_{\gamma_{n,k}}(x)$ counts the number of directed paths of length 2 from $x$ to $x_l$, and $\text{Right}_{\gamma_{n,k}}(x)$ analogously.

We conjecture that the difference $\text{Right}_{\gamma_{n,k}} - \text{Left}_{\gamma_{n,k}}$ can be used as before to construct a density estimate. Specifically, if $\gamma_{n,k}$ is a shortest path from the anchor point $X_0$ to $X_s$, we believe that under similar conditions on $k$ and $n$ as before,

$$\frac{\eta_d}{k\nu_d} \sum_{x \in \gamma_n} \text{Right}_{\gamma_{n,k}}(x) - \text{Left}_{\gamma_{n,k}}(x) \tag{$\star$}$$

is a consistent estimator of the quantity $\log p(X_s) - \log p(X_0)$. Our simulations in Section 6 show that the estimate works, even surprisingly well. So far we do not have a formal proof yet, due to two technical difficulties. The first problem is that the set $\operatorname{In}(x_l)$ is not a ball, but an "egg-shaped" set. As $n \to \infty$, one can sandwich $\operatorname{In}(x)$ between two concentric balls that converge to each other, but this approximation is too weak to carry the proof. To compute the expected value $E(\text{Right}_{\gamma_{n,k}}(x) - \text{Left}_{\gamma_{n,k}}(x))$ we would have to integrate the intersection of the "egg" $\operatorname{In}(x_l)$ with the ball $\operatorname{Out}(x)$, and so far we have no closed form solution. The second difficulty is related to the shortest path in the graph. While it is clear that "most edges" in this path have approximately the maximal length (that is, $(k/(n\eta_d p(x)))^{1/d}$ for an edge in the neighborhood of $x$), this is not true for all edges. Intuitively it is clear that the contribution of the few violating edges will be washed out in the integral along the shortest path, but we don't have a formal proof yet.

What we can prove is the following weaker version. Consider a $D_p$-shortest path $\gamma \subset \mathbb{R}^d$ and a point $x$ on this path with out-radius $r_{out}(x)$. Define the points $x_l$ and $x_r$ as the two points where the path $\gamma$ enters resp. leaves the ball $B(x, r_{out}(x))$, and define the sets $L_{n,k} := \operatorname{Out}(x) \cap B(x_l, r_{out}(x))$ and $R_{n,k} := \operatorname{Out}(x) \cap B(x_r, r_{out}(x))$. Then it can be proved that $(\eta_d^{1/d})/(k\nu_d^{1/d}) \sum_{x \in \gamma_n} R_{n,k}(x) - L_{n,k}(x) \to \log p(X_s) - \log p(X_0)$. The proof is similar to the one in Section 4. It circumvents the problems mentioned above by using well defined balls instead of In-sets and the continuous path $\gamma$ rather than the finite sample shortest path $\gamma_n$, but the quantities cannot be estimated from the kNN graph alone.

# 6 Simulations

As a proof of concept, we ran simple experiments to evaluate the behavior of estimator $(\star)$. We draw $n = 2000$ points according to a couple of simple densities on $\mathbb{R}$, $\mathbb{R}^2$ and $\mathbb{R}^{10}$, then we build the directed, unweighted kNN graph with $k = 50$. We fix a random point as anchor point $X_0$, compute the quantities $\mathrm{Right}_{\gamma_{n,k}}$ and $\mathrm{Left}_{\gamma_{n,k}}$ for all sample points, and then sum the differences $\mathrm{Right}_{\gamma_{n,k}} - \mathrm{Left}_{\gamma_{n,k}}$ along shortest paths to $X_0$. Rescaling by the constant $\eta_n/(kv_d)$ and exponentiating then leads to our estimate for $p(x)/p(X_0)$. In order to nicely plot our results, we multiply the resulting estimate by $p(X_0)$ to get rid of the scaling constant (this step would not be possible in applications, but it merely serves for illustration purposes). The results are shown in Figure 3. It is obvious from these figures that our estimate "works", even surprisingly well (note that the sample size is not very large and we did not perform any parameter tuning). Even in the case of a step function the estimate recovers the structure of the density. Note that this is a particularly difficult case in our setting, because within the constant parts of the two steps, the kNN graphs of the left and right step are indistinguishable. It is only in a small strip around the boundary between the two steps that kNN graph will reveal non-uniform behavior. The simulations show that this is already enough to reveal the overall structure of the step function.

# 7 Extensions

We have seen how to estimate the density in an unweighted, directed kNN graph. It is even possible to extend this result to more general cases. Here is a sketch of the main ideas.

**Estimating the dimension from the graph.** The current density estimate requires that we know the dimension $d$ of the underlying space because we need to be able to compute the constants $\eta_d$ (volume of the unit ball) and $v_d$ (intersection of two unit balls). The dimension can be estimated from the directed, unweighted kNN graph as follows. Denote by $r$ the distance of $x$ to its $k$th-nearest neighbor, and by $K$ the number of vertices that can be reached from $x$ by a directed shortest path of length 2. Then $k/n \approx P(B(x,r))$ and $K/n \approx P(B(x,2r))$. If $n$ is large enough and $k$ small enough, the density on these balls is approximately constant, which implies $K/k \approx 2^d$ where $d$ is the dimension of the underlying space.

**Recovering the directed graph from the undirected one.** The current estimate is based on the directed kNN graph, but many applications use undirected kNN graphs. However, it is possible to recover the directed, unweighted kNN graph from the undirected, unweighted graph. Denote by $N(x)$ the vertices that have an undirected edge to $x$. If $n$ is large and $k$ small, then for any two vertices $x$ and $y$ we can approximate $|N(x) \cap N(y)|/n \approx P(B(x,r) \cap B(y,r))$. The latter is monotonously decreasing with $\|x - y\|$. To estimate the set $\mathrm{Out}(x)$ in order to recover the directed kNN graph, we rank all points $y \in N(x)$ according to $|N(x) \cap N(y)|$ and choose $\mathrm{Out}(x)$ as the first $k$ vertices in this ranking.

**Point embedding.** In this paper we focus on estimating the density from the unweighted kNN graph. Another interesting problem is to recover an embedding of the vertices to $\mathbb{R}^d$ such that the kNN graph based on the embedded vertices corresponds to the given kNN graph. This problem is closely related to a classic problem in statistics, namely non-metric multidimensional scaling (Shepard, 1966, Borg and Groenen, 2005), and more specifically to learning distances and embeddings from ranking and comparison data (Schultz and Joachims, 2004, Agarwal et al., 2007, Ouyang and Gray, 2008, McFee and Lanckriet, 2009, Shaw and Jebara, 2009, Shaw et al., 2011, Jamieson and Nowak, 2011) as well as to ordinal (monotone) embeddings (Bilu and Linial, 2005, Alon et al., 2008, Bădoiu et al., 2008, Gutin et al., 2009). However, we are not aware of any approach in the literature that can faithfully embed unweighted kNN graphs and comes with performance guarantees. Based on our density estimate, such an embedding can now easily be constructed. Given the unweighted kNN graph, we assign edge weights $w(X_i, X_j) = (\hat{p}^{-1/d}(X_i) + \hat{p}^{-1/d}(X_j))/2$ where $\hat{p}$ is the estimate of the underlying density. Then the shortest paths in this weighted kNN graph converge to the Euclidean distances in the underlying space, and standard metric multidimensional scaling can be used to construct an appropriate embedding. In the limit of $n \to \infty$, this approach is going to recover the original point embedding up to similarity transformations (translation, rotation or rescaling).

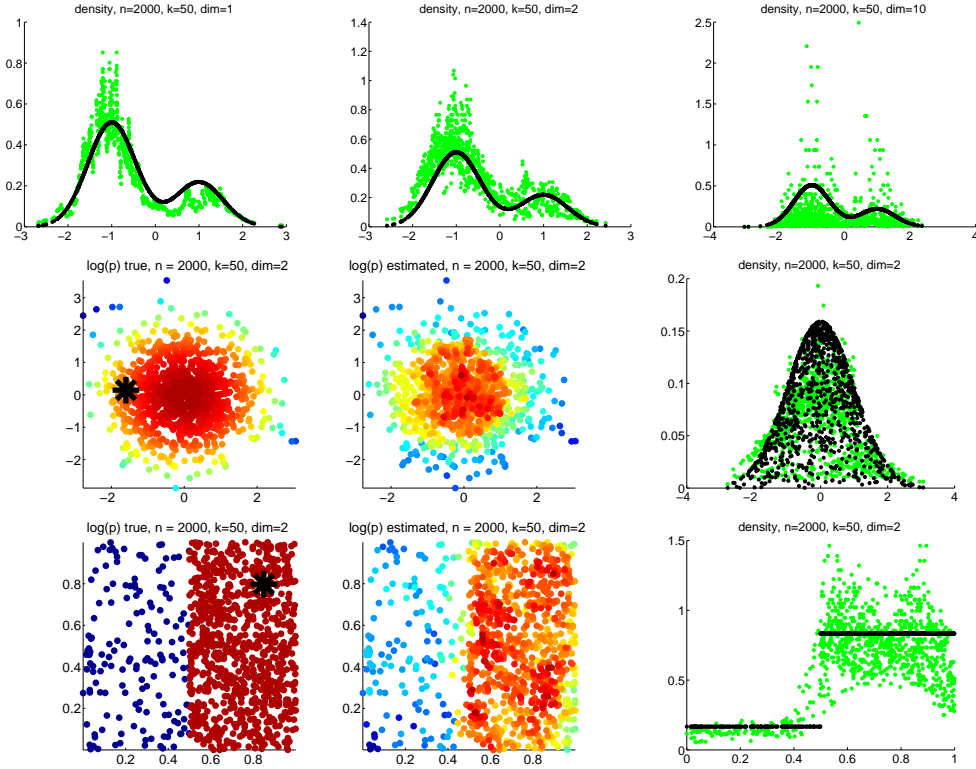

Figure 3: Densities and their estimates. Density model in the first row: the first dimension is sampled from a mixture of Gaussians, the other dimensions from a uniform distribution. The figures plot the first dimension of the data points versus the true (black) and estimated (green) density values. From left to right, they show the case of 1, 2, and 10 dimensions, respectively. Second and third row: 2-dimensional densities. The left plots show the true log-density (a Gaussian and a step function), the middle plots show the estimated log-density. The right figures plot the first coordinate of the data points against the true (black) and estimated (green) density values. The black star in the left plot depicts the anchor point $X_0$ of the integration step.

## 8 Conclusions

In this paper we show how a density can be estimated from the adjacency matrix of an unweighted, directed kNN graph, provided the graph is dense enough ($k^{d+2}/(n^2 \log^d n) \to \infty$). In this case, the information about the underlying density is implicitly contained in unweighted kNN graphs, and, at least in principle, accessible by machine learning algorithms. However, in most applications, $k$ is chosen much, much smaller, typically on the order $k \approx \log(n)$. For such sparse graphs, our density estimate fails because it is dominated by sampling noise that does not disappear as $n \to \infty$. This raises the question whether this failure is just an artifact of our particular construction or of our proof, or whether a similar phenomenon is true more generally. If yes, then machine learning algorithms on sparse unweighted kNN graphs would be highly problematic: If the information about the underlying density is not present in the graph, it is hard to imagine how machine learning algorithms (for example, spectral clustering) could still be statistically consistent. General lower bounds proving or disproving these speculations are an interesting open problem.

**Acknowledgements**

We would like to thank Gabor Lugosi for help with the proof of Theorem 1. This research was partly supported by the German Research Foundation (grant LU1718/1-1 and Research Unit 1735 "Structural Inference in Statistics: Adaptation and Efficiency").

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
