[Reviews · NeurIPS 2013]

Submitted by Assigned_Reviewer_1

The paper proposes a new knn density estimation that uses only the unweighted knn adjacency graph.

The paper is readable, interesting, and the topic is important.

The main result is Theorem 5, which claims the consistency of the proposed estimator.
I really like Section 3, which explains nicely the main ideas of the paper in the simplest 1d case.

Weak points of the paper:
(i) It would have been nice to see results about the convergence rate of the estimators too, but unfortunately the authors didn’t study this question.
(ii) The proof is only a sketch, and technical details are missing.
(iii) The simulations in the paper are only meant to be a proof of concept. It would have been nice to see numerical experiments on real data where the proposed approach can work well, but it is difficult to use other density estimators. These experiments on real datasets could also motivate the paper better.


Summary: It’s an interesting paper about a surprising result. The motivation could be improved, and numerical experiments on real datasets could demonstrate the applicability of the proposed method.

Submitted by Assigned_Reviewer_4

QUALITY:
This work shows an interesting way of estimating the density of a random vector from an unweighted k-NN graph. It assumes that the density is differentiable and bounded away from zero in the epsilon-interior of the support.

Some minor points:
Page 2: in the definition of In(x), "E" has not been defined. Did the authors mean "E_n"?
Page 2: last paragraph: the unweighted shortest path distance needs to be defined.
Page 4: Proposition 1: I guess the condition n/log(k) goes to infinity must be changed to k/log(n) goes to infinity.


CLARITY:
The paper is nicely written. The main results are motivated with examples and illustrations, and the proofs of the results are sufficiently sketched.


ORIGINALITY:
The main idea is to use rectifiable paths connecting two vertices to define "order" (that is right and left) in a higher dimensional space. This is a clever idea. It is known that in Euclidean space, a weighted k-NN graph can give a rough (but asymptotically acceptable) estimate of the density. In this work, the authors conjecture that even the unweighted k-NN graph gives an estimate of the density. The authors show support for their conjecture by showing that asymptotically their procedure converges to the true density with a little more knowledge of the geometry. But as in the case of the weighted graph, for real life applications, this estimate may not be too good, unless the size of the sample is large. This is what the authors do in their simulation: n=2000 with k=50 in one or two dimensions. Higher dimensions will require an enormous quantity if data. The procedure uses the knowledge of the vector dimension. In an unweighted graph, the actual dimension may be unknown. The suggestion given by the authors to estimate it will work again only if n is very large.


SIGNIFICANCE:
I like the extended application suggested at the end of the paper for multi-dimensional scaling. This might be a good and very practical application of the procedure.

Summary: Density estimation from a k-NN unweighted graph is suggested and conjectured to give a consistent estimate for large data size and k. The paper is based on clever geometric ideas. The results and suggestions given in the paper, if proven correct, could be used widely in applications.

Submitted by Assigned_Reviewer_5

A method of estimating a density (up to constants) from an unweighted, directed k nearest neighbor graph is described. It is assumed (more or less) that the density is continuously differentiable, supported on a compact and connected subset of R^d with non-empty interior and a smooth boundary, and is upper- and lower-bounded on its support.

The method is based on two ideas. The first is to use shortest paths in the graph as a way to estimate the "orientation" of vertices in the original space, i.e. to find pairs of groups of points that, in the original embedding, are nearby, non-intersecting, and aligned with the curve that corresponds to the shortest density-weighted path in R^d (between the same endpoints).

The second idea is to integrate a quantity that can be estimated from the sets described above over the shortest paths to find the log likelihood ratio between the start and end points of the path.

The paper provides an interesting perspective on graph-based supervised and unsupervised machine learning methods, as well as their connection to a class of density-based distances that has been of interest in the community recently. Though likely of no immediate practical importance, the reviewer believes that there may be significant theoretical insights to be gained from this work.

The proof of consistency for the version of the proposed method that depends only on the k-NN graph is incomplete, as it depends on a technical conjecture. However, it is not difficult to believe that the conjecture, or at least a similar statement, holds.

Minor points:

It is worth mentioning that the proposed method for estimating the dimension from the graph in section 6 is well known in the data mining community.

In proposition 1, n/\log k \rightarrow \infty probably should read k/\log n instead.
Summary: A method of estimating a density (up to constants) from an unweighted, directed k nearest neighbor graph is described. Though likely of no immediate practical importance, the reviewer believes that there may be significant theoretical insights to be gained from this work.
Author Feedback

Author rebuttal: We thank the reviewers for their time, effort and comments! We will make sure to implement the suggested changes.